# Are the Yellow and Red Marked Club-Tail *Losaria coon* the Same Species?

**DOI:** 10.3390/insects11060392

**Published:** 2020-06-24

**Authors:** Zhen-Bang Xu, Yun-Yu Wang, Fabien L. Condamine, Adam M. Cotton, Shao-Ji Hu

**Affiliations:** 1School of Agriculture, Yunnan University, Kunming 650500, China; zhenbangxu@mail.ynu.edu.cn; 2Plant Protection College, Yunnan Agricultural University, Kunming 650201, China; wangyunyu@mail.kiz.ac.cn; 3CNRS, Institut des Sciences de l’Evolution de Montpellier (Université de Montpellier|CNRS|IRD|EPHE), Place Eugène Bataillon, 34095 Montpellier, France; fabien.condamine@gmail.com; 486/2 Moo 5, Tambon Nong Kwai, Hang Dong, Chiang Mai 50230, Thailand; adamcot@cscoms.com; 5Yunnan Key Laboratory of International Rivers and Transboundary Eco-security, Yunnan University, Kunming 650500, China; 6Institute of International Rivers and Eco-Security, Yunnan University, Kunming 650500, China

**Keywords:** species delimitation, distribution range, molecular phylogeny, wing pattern, genitalic structure

## Abstract

*Losaria coon* (Fabricius, 1793) is currently comprised of ten subspecies, which were originally described under two names, *Papilio coon* and *P. doubledayi* before 1909, when they were combined as one species. The main difference between them is the colour of abdomen and hindwing subterminal spots—yellow in *coon* and red in *doubledayi*. Wing morphology, male and female genitalia, and molecular evidence (DNA barcodes) were analysed for multiple subspecies of *L. coon* and three other *Losaria* species—*rhodifer*, *neptunus*, and *palu*. Our molecular data support the separation of *L. coon* and *L. doubledayi*
**stat. rev.** as two distinct species, with *L. rhodifer* positioned between them in phylogenetic analyses. Wing morphology and genitalic structures also confirm the molecular conclusions. Our findings divide *L. coon* into two species occupying different geographic ranges: with *L. coon* restricted to southern Sumatra, Java, and Bawean Island, while *L. doubledayi* occurs widely in regions from North India to northern Sumatra, including Hainan and Nicobar Islands. Hence, future conservation efforts must reassess the status and threat factors of the two species to form updated strategies.

## 1. Introduction

*Losaria coon* (Fabricius, 1793) is a club-tail papilionid butterfly of the genus *Losaria* (Moore, 1902), which contains three other species, namely *L. rhodifer* (Butler, 1876), *L. neptunus* (Guérin-Méneville, 1840), and *L. palu* (Martin, 1912) [1]. *L. coon* is distributed across the Oriental tropics, ranging from northern India to Java, including Indochina, southern China, and Sumatra [2].

The species currently recognised as *Losaria coon* contains a number of taxa, some of which were originally described as species or as subspecies of the yellow or red marked species, *coon* and *doubledayi,* respectively, which were originally described in the genus *Papilio* (Linnaeus, 1758) [3,4]. All 19th century publications (e.g., Rothschild [5]) treated *P. coon* and *P. doubledayi* as separate species, as did Moore [6]. Jordan [7], without explanation united them as a single species, *Papilio coon*, which has been followed by all subsequent publications (e.g., Evans [8] and Talbot [9]). Modern works have placed the species in other genera than *Papilio*, mostly treating it in the genus *Atrophaneura* (Reakirt, 1865), following Ford [10]. Tsukada and Nishiyama [2] and Hancock [1] separated the club-tailed species in *Losaria* (Moore, 1902), where they are currently placed. This genus designation is also supported by the current molecular phylogenies of the family Papilionidae [11].

To date, ten subspecies have been placed under *L. coon* [2,12,13,14], which can be divided into two groups with the following morphological characters and geographic distributions, as shown in Figure 1: (a) four subspecies occupying western and northern Java, southwest Sumatra, and Bawean Island, respectively, ssp. *palembanganus* (Rothschild, 1895), ssp. *coon*, ssp. *patianus* (Fruhstorfer, 1898), and ssp. *sangkapurae* (Bollino and Sala, 1992), possessing yellow marginal spots on the hindwing (referred to as yellow marked *coon*) with the hindwing discal cell patch reaching the base of the cell; (b) six subspecies found from northeastern India, northern Myanmar, Nicobar Islands, Indochina and Malay Peninsula, Hainan Island, and northern Sumatra, namely ssp. *cacharensis* (Butler, 1885), ssp. *putaoa* (Tytler, 1939), ssp. *sambilanga* (Doherty, 1886), ssp. *doubledayi* (Wallace, 1865), ssp. *insperatus* (Joicey and Talbot, 1921), and ssp. *delianus* (Fruhstorfer, 1895), which all share red marginal spots on the hindwing (referred to as red marked *coon*), and the hindwing discal cell patch does not reach the base of the cell.

The yellow and red marked *coon* are separated from each other geographically, even in Sumatra, where the yellow marked ssp. *palembanganus* and the red marked ssp. *delianus* are not sympatric, according to available records, as shown in Figure 1. The constant morphological differences coupled with this distribution pattern leave an unanswered question—whether the yellow and red marked *coon* really belong to the same species.

Speciation with limited morphological differentiation among closely related taxa is not unique to *L. coon* in Lepidoptera. Some recent studies revealed that speciation caused by recent vicariance produced multiple morphologically similar taxa which have long been treated as subspecies of the same species (e.g., the dragontail butterflies of genus *Lamproptera* (Gray, 1832) [15], the swordtail butterflies of subgenus *Pazala* (Moore, 1888) [16,17,18], the Asian moon moths of genus *Actias* (Leach, 1815) [19], the hawkmoths of genus *Cechetra* (Zolotuhin and Ryabov, 2012) [20], and genus *Laothoe* (Fabricius, 1807) [21]). Apart from traditional morphological comparisons, such as wing pattern and genitalia analyses, the abovementioned studies applied mitochondrial DNA analyses (mostly barcode sequences) [22] to facilitate elucidating some long-lasting taxonomic confusions.

The aim of the present study is to revise the taxonomic identities of the yellow and red marked *coon* using mitochondrial DNA barcode sequences combined with morphological comparisons. Our findings will first clarify taxonomic confusions, draw updated geographic ranges for relevant taxa, and finally benefit from the formulation of better conservation strategies [12].

## 2. Materials and Methods

### 2.1. Taxon Sampling

Specimens of *L. coon* (including four subspecies), *L. rhodifer* (Butler, 1876), *L. neptunus* (Guérin-Méneville, 1840) (including two subspecies), and *L. palu* (Martin, 1912) in this study were mainly sampled from the authors’ private collections. Subspecies identification for *Losaria* species mainly followed Tsukada and Nishiyama [2]. Specimens were collected and dried in paper triangles at room temperature until use. For each individual used in molecular analysis, two legs from the same side were taken for DNA extraction before the specimen was rehydrated for spreading. An individual of *Pharmacophagus antenor* (Drury, 1773) and an individual of *Pachliopta polyphontes* (Boisduval, 1836) were chosen as outgroups for phylogenetic analyses, similar to those used by Hancock [1]. In addition, three sequences, including two individuals of *L. coon doubledayi* and an individual of *L. neptunus neptunus*, were mined from the Barcode of Life Database v.4 (BOLD) (http://www.boldsystems.org) for the molecular analyses. The collected data and GenBank accession numbers of all samples and mined sequences used in our molecular analyses are listed in Table 1.

Due to difficulties obtaining fresh samples of some Indian, Myanmar, and Indonesian insular subspecies, namely *L. coon cacharensis* (Butler, 1885), *L. coon sambilanga* (Doherty, 1886), *L. coon putaoa* (Tytler, 1939), *L. coon delianus* (Fruhstorfer, 1895), *L. coon patianus* (Fruhstorfer, 1898), and *L. coon palembanganus* (Rothschild, 1896), these taxa are not included in the molecular analysis of this study, but will be discussed based on morphological characters from photos of type specimens and original descriptions (Appendix A).

### 2.2. DNA Extraction and Amplification

The phenol–chloroform protocol was used to extract genomic DNA from two legs pulled from the same side of a specimen of all sampled taxa. The legs were homogenised in protease buffer containing 450 μL STE (10mmol/L Tris-HCl, 1 mmol/L EDTA, 100 mmol/L NaCl, pH = 8.0), 25 μL Proteinase K (20 mg/mL) and 75 μL SDS (10%) and incubated at 55 °C for 12 h to rehydrate and lyse the tissue. The subsequent extraction protocol followed that reported by Hu et al. [23]. The resultant genomic DNA was preserved at −40 °C.

The polymerase chain reaction (PCR) was carried out in a 25 μL system by using the TaKaRa Ex *Taq* Kit (TaKaRa Biotechnology Co., Ltd., Dalian, China). The system contained 2.5 μL 10× PCR buffer, 2.0 μL MgCl_2_ (25 mmol/L), 2.0 μL dNTP mixture (2.5 mmol/L each), 0.25 μL *Taq* DNA polymerase (5 U/μL), and 0.5 μL each of forward and reverse primers (20 μmol/L). The mitochondrial *cox1* gene fragment (the DNA barcode) was amplified and sequenced with the following primers LCO1490 (5′- GGT CAA CAA ATC ATA AAG ATA TTG G-3′) and HCO2198 (5′- TAA ACT TCA GGG TGA CCA AAA AAT CA-3′) [24]. The thermal profile of PCR consisted of an initial denaturation at 95 °C for 3 min, 30 cycles of denaturation at 94 °C for 1 min, annealing at 50 °C for 1 min, and elongation at 72 °C for 1 min; then a final elongation at 72 °C for 5 min. The sequences were obtained using an ABI Prism 3730 sequencer (Applied Biosystems, Foster City, CA, USA).

### 2.3. Phylogenetic Analyses

We proofread and aligned the raw sequences with Clustal W [25] in BioEdit 7.0.9 [26] by examining the chromatograms to ascertain polymorphic sites, and problematic sequences were excluded when double peaks were present in the chromatograms. MEGABLAST was applied to check the identities of all sequences against the genomic references and nucleotide collections in GenBank, and amino acid translation was realised with the invertebrate mitochondrial criterion in MEGA 7.0 [27] to detect possible *Numts* (nuclear copies of mtDNA fragments). A search for non-synonymous mutations in-frame stop codons and indels was carried out to further minimise the existence of cryptic *Numts* [28,29]. The Kimura two-parameter distances [30] between taxa were calculated in MEGA 7.0.

All sequences were used in phylogenetic reconstructions without pruning identical haplotypes, since the monophyly of samples identified as species by morphological characters [2,13,14] needs to be tested. The phylogeny was reconstructed using a Bayesian Inference (BI) as implemented in MrBayes 3.2.6 [31], with the most appropriate partition scheme recovered by PartitionFinder 2.1.1 [32] using the unlinked branch lengths and the *greedy* algorithm. We used the partitioning scheme among site rate variation, suggested by PartitionFinder, but instead of selecting one substitution model a priori, we used reversible-jump Markov Chain Monte Carlo (rj-MCMC) to allow sampling across the entire substitution rate model space [33]. BI analyses consisted of two independent runs, each with eight rj-MCMC running for five million generations (sampled every 1000th generation) to calculate the clade posterior probabilities (PP). The marginal likelihood estimate was performed with the stepping-stone sampling [34], implemented in MrBayes with 100 steps, each with 10 million generations, and a diagnostic frequency of 1000. Through the computation of Bayes factors, marginal likelihood estimates were used to compare the model fit of an unconstrained topology with the fit of a constrained topology, in which a group of specimens is forced to be monophyletic.

### 2.4. Molecular Species Delimitation

In order to test the delimitation of *Losaria* species, we relied on a tree-based approach with the Bayesian–Poisson tree process model (bPTP; https://species.h-its.org/ptp/) [35], and on a DNA-based approach with the Automatic Barcode Gap Discovery (ABGD; https://bioinfo.mnhn.fr/abi/public/abgd) [36]. The bPTP model estimates the probability of each clade being a putative species based on the branch lengths of a tree. The bPTP analyses were performed with the guide tree, reconstructed from the BI in MrBayes with the following parameters: 100,000 MCMC generations, thinning every 100 generations, 0.1% of generation discarded as burn-in. In comparison, the ABGD model estimates the barcode gap separated the intraspecific from the interspecific molecular divergence. Thus, the prior maximum divergence of intraspecific diversity (*P*) is an important component of ABGD as it provides approximate indications on the barcode gap. If *P* is set too high, the whole dataset will be considered as a single species, while if *P* is set too low, only identical sequences will be considered as part of the same species. Previous results showed that the number of species ranges from 1 (generally when *P* = 0.1) to a large number of species that corresponds to groups of identical sequences (generally when *P* = 0.001). We followed this practice and set the range of *P* from 0.001 to 0.1 to explore a lumped versus split delimitation. In addition, the Kimura two-parameter distance was selected but the analyses made with the Jukes–Cantor model provided identical results. The sensitivity of the method to gap width was left by default (1.5), but analyses performed with higher gap width (2 and 3) yielded identical results. The remaining parameters were left by default. Finally, the Monophylizer (http://monophylizer.naturalis.nl/) [37] was performed on the obtained BI tree to test the monophyly of each identified taxon.

### 2.5. Morphological Comparison

Specimens were spread for examination, with the anal scent scales exposed. Species identification was performed prior to molecular work. All spread specimens were photographed using a digital camera, and Adobe Photoshop CS (Adobe, San Jose, CA, USA) was used to adjust the exposure of these photos.

The methods for preparing male and female genitalia followed Hu, Cotton, Condamine, Duan, Wang, Hsu, Zhang, and Cao [16]. The abdomen was removed from the specimen and placed into a 1.5 mL microcentrifuge tube, and 1 mL water was added to rehydrate the tissue at 50 °C for 30 min, then 1 mL 10% sodium hydroxide solution was used to digest soft tissue at 70 °C for 1h. The treated abdomen was neutralised with 2% acetic acid and then dissected in a water-filled Petri dish under the stereoscope to remove residual tissues, scales, and hair. The genitalia were then transferred to 80% glycerol for 12 h to render them transparent. Photographs were taken with a Nikon DMX1200 digital camera (Nikon, Japan), mounted on a Nikon SMZ1500 stereoscope (Nikon, Japan) and automatically stacked using Helicon Focus 7 (Helicon Software, Kharkiv, Ukraine). All parts of the genitalia were fixed on a glue card with water-soluble white glue and pinned with the specimen after observation and photography.

## 3. Results

### 3.1. Molecular Phylogenetic Relationships

Bayesian phylogenetic analyses converged well, as indicated by the average standard deviation of split frequencies close to 0 (0.003864), potential scale reduction factors equal to 1 (maximum = 1.003), and effective sample size >> 200 for all parameters. The phylogeny of the *Losaria* species was recovered as two major clades with maximal PP value, as shown in Figure 2. The clade (PP = 1) containing both the yellow and red marked *coon* and *L. rhodifer* is younger than the other clade including *L. neptunus* and *L. palu* (PP = 1).

In the first major clade, the yellow and red marked *coon* is divided into two monophyletic subclades by *L. rhodifer* (PP = 0.83), indicating a specific level divergence between the two morphological types. The subclade of yellow marked *coon* contains *L. coon coon* and *L. coon sangkapurae*, and that of red marked *coon* contains *L. coon insperatus* and *L. coon doubledayi*, which were further split into two paraphyletic branches, as shown in Figure 2. The basal branch includes samples collected from the northern Malay Peninsula (not far from the type locality of *doubledayi*), and the sister branch includes samples from Indochina, implying a certain degree of divergence between the two geographical populations, as shown in Figure 2.

The estimations of marginal likelihoods with stepping-stone sampling in MrBayes confirmed the non-monophyly of *Losaria coon* when all specimens (*coon coon*, *coon sangkapurae*, *coon doubledayi*, and *coon insperatus*) are constrained to form a monophyletic group (logL = −1955.68 for the unconstrained analysis, logL = −1960.01 for the constrained analysis and Bayes factors = 8.66).

### 3.2. Molecular Species Delimitation

Both bPTP and ABGD approaches produced similar results of taxon delimitation. They identified five *Losaria* species and four subspecific taxa from the Bayesian tree, excluding two outgroups. The bPTP model identified the yellow and red marked *coon* as separate species with a probability of 0.81 for *L. coon doubledayi* and of 0.89 for *L. coon coon*, and with probabilities above 0.65 for *L. rhodifer* (probability = 0.98) *L. neptunus* (probability = 0.76) and *L. palu* (probability = 0.87), as shown in Figure 2. For subspecies, the bPTP model identified subspecies of the red marked *coon* with supporting values ranging from 0.66 to 0.81, whereas a combination of *L. coon sangkapurae* with *L. coon coon* produced a very low supporting value of 0.03, as shown in Figure 2. The ABGD approach also identified the yellow and red marked *L. coon* as distinct species in all the analyses, whatever the settings for the nucleotide model, the gap width, and the range of *P*. At a prior intraspecific divergence *P* > 0.008, we inferred five groups of sequences corresponding to the five *Losaria* species. Four subspecific taxa are recovered at *P* > 0.002. ABGD lumps all *coon coon*, *coon sangkapurae*, *coon doubledayi*, *coon insperatus*, and *rhodifer* specimens into a single species for a prior intraspecific divergence *P* > 0.02.

The monophylizer analysis also identified the yellow and red marked *coon* as two monophyletic clades in the BI tree, along with *L. rhodifer*, *L. neptunus*, and *L. palu*. Paraphyletic clades were found in subspecies, such as *L. coon coon* (one of the yellow marked *coon*) and the two *doubledayi* races (red marked *coon*) from southern Thailand to the Malay Peninsula and Indochina, as shown in Table 2.

The Kimura two-parameter (K2P) distances between the yellow and red marked *coon* reached 2.95%. The K2P distances between all taxa within both groups, and the species of *L. rhodifer*, *L. neptunus*, and *L. palu* ranged from 0.19% to 9.61%, with that between *L. coon coon* and *L. coon sangkapurae* being the smallest, and that between *L. rhodifer* and *L. palu* being the greatest, as shown in Table 3. For those distances between the yellow and red marked *coon*, the overall range fell from 2.44% to 3.17%, which exceeds the proposed barcoding gap of 2% [38], while the K2P distances within the yellow or red marked *coon,* respectively, remained at infraspecific level (0.19–1.08%), as shown in Table 3. It is noteworthy that the K2P distances between the yellow marked *coon* and *L. coon insperatus* (one of the red marked subspecies) are greater (3.57–3.77%) than those between the yellow marked *coon* and *L. rhodifer* (3.17–3.35%), and the distances between the yellow marked *coon* and the Malay Peninsular *doubledayi* are very close to those between the yellow marked *coon* and *L. rhodifer* (3.08–3.28% vs. 3.17–3.35%), as shown in Table 3.

Given the aforementioned molecular evidence, we propose that the yellow and red marked *coon* belong to two distinct species, namely *Losaria coon* (Fabricius, 1793) and *Losaria doubledayi* (Wallace, 1865) **stat. rev.**, respectively. The subsequent morphological analyses and discussion will be based on the separated names.

### 3.3. Morphological Differences

Comparison of the wing morphology between *L. coon* and *L. doubledayi* showed constant differences, as illustrated in Figure 3. Beside the evident yellow and red colouration on the abdomens and hindwings of both species, the following characters are also important in separating the two species: (1) the hindwing white discal cell patch reaches the wing base in *L. coon*, but only reaches about half way to the base in *L. doubledayi* (a); (2) the forewing ground colour in *L. coon* is buffish brown, whereas in *L. doubledayi* it is blackish (b); the colour of the male scent scales is light buffish in *L. coon*, but greyish black in *L. doubledayi* (f); (3) the subterminal spot in the cell M_2_ of the hindwing is usually larger in *L. coon* than in *L. doubledayi* (c); (4) the junction between the termen and the lobe at the end of vein CuA_1_ on the hindwing is evidently angled in *L. coon*, but much straighter in *L. doubledayi* (d); (5) in the hindwing tail, the stalk is broader in *L. coon* than that in *L. doubledayi*, and the club is narrower in *L. coon* than that in *L. doubledayi* (e).

In overall appearance, among the remaining three *Losaria* species, *L. rhodifer* is more similar to *L. coon* and *L. doubledayi*. However, the more fragmented hindwing white patch, much larger subterminal crimson spots, and the iconic crimson-tipped tail are ready separation characters, as shown in Figure 4. Morphological similarity and differences of *L. rhodifer* agrees with its phylogenetic position obtained by our molecular analyses, as shown in Figure 2. The other two species, *L. neptunus* and *L. palu*, are less closely related to the previously mentioned taxa, judging from the distinctive yellow-tipped abdomens and lack of hindwing discal patches, as shown in Figure 4.

Male genitalia of available subspecies were dissected, constant differences were found between *L. coon* and *L. doubledayi*, while the characters remain consistent within each group, as shown in Figure 5. The main differences in male genitalia are: (1) the dorsal view of the superuncus is broader in *L. coon* (0.29 ± 0.03 mm, *n* = 12), while it is rather slender (0.16 ± 0.02 mm, *n* = 15) in *L. doubledayi*; (2) the tip of the valve is only truncated in *L. coon* but strongly indented in *L. doubledayi*, forming a bifid tip; (3) the apical half of the juxta is 1.3 to 1.5 times broader in *L. coon* than in *L. doubledayi*.

Female genitalia of available subspecies were dissected, constant differences were found between *L. coon* and *L. doubledayi*, while the characters remain consistent within each group, as shown in Figure 6. The main differences in female genitalia are: (1) the ostium is smaller in *L. coon* but broader in *L. doubledayi*; (2) the lamella antevaginalis sunk inwardly, forming a crescent edge, in *L. coon,* while forming an even edge in *L. doubledayi*; (3) the lamella postvaginalis is 0.5 to 0.7 times shorter and 1.8 to 2.0 times broader in *L. coon* than that in *L. doubledayi*; (4) the lateral area surrounding the ostium is less developed in *L. coon* than in *L. doubledayi*; (5) the signum is leaf-shaped with one end longer than the other in *L. coon,* while nearly round in *L. doubledayi*, and the length of the signum in *L. coon* is 1.7 to 2.0 times longer than that in *L. doubledayi*.

Among the male and female genitalia of the other three species of *Losaria*, those of *L. rhodifer* are similar to the previously mentioned two species, but the shape of valve tip of the male genitalia and the ostium, lamella antevaginalis, and signum are still different, as shown in Figure 7A and Figure 8A,D. Such resemblance indicates that *L. rhodifer* is closely related to both *L. coon* and *L. doubledayi,* as previously mentioned. The genitalia of both sexes of *L. neptunus* and *L. palu* are all significantly different, as shown in Figure 7B,C and Figure 8B–F, supporting the phylogenetic relationships with *L. coon* and *L. rhodifer,* as inferred by the molecular data.

### 3.4. Updated Subspecies Checklist of Losaria (Moore, 1902)


***Losaria* (Moore, 1902)**


*Losaria* (Moore, 1902); Lepidoptera Indica, **5**(57): 184; TS: *Papilio coon* (Fabricius, 1793).

  *Balignina* (Moore, 1902); Lepidoptera Indica, **5**(57): 187; TS: *Papilio neptunus* (Guérin-Méneville, 1840).


***Losaria palu* (Martin, 1912)**


*Papilio palu* (Martin, 1912); D. ent. Z. Iris, **26**(3): 164; TL: “Lewara ... 2000 m ... Palu” [Lewara, Palu, Sulawesi, Indonesia].

  **Distribution:** Only known from the vicinity of Palu, Sulawesi.


***Losaria neptunus* (Guérin-Méneville, 1840)**


  ***L. neptunus manasukkiti* (Cotton, Racheli and Sukhumalind, 2005)**


  *Losaria neptunus manasukkiti* (Cotton, Racheli and Sukhumalind, 2005); Fragm. ent., **37**(1): 130, f. 1-2; TL “Kamphuan, Ranong, Thailand”.

  **Distribution:** Ranong Province, Thailand and southernmost Myanmar.

  ***L. neptunus neptunus* (Guérin-Méneville, 1840)**


  *Papilio neptunus* (Guérin-Méneville, 1840); Revue zool., **3**(2): 43; TL: “Côte Malaye” [Malay Peninsula].

     *Papilio thetys* (Guenée, 1872); Mém. Soc. Phys. Hist. nat. Genève **21**(2): 378 pl. f. 5; TL: “pas d’indication de localité” [without indication of locality].

  **Distribution:** West Malaysia, southernmost Thailand.

  ***L. neptunus creber* (van Eecke, 1914)**

  *Papilio neptunus* ssp. *creber* (van Eecke, 1914); Notes Leyden Mus., **36**(3/4): 197; TL: “Sinabang; Sibigo” [Sinabang and Sibigo, Simeulue Is., Indonesia].

  **Distribution:** Simeulue Is. [Indonesia].

  ***L. neptunus fehri* (Honrath, 1892)**

  *Papilio neptunus* var. *fehri* (Honrath, 1892); Berl. Ent. Z., **36**(2): 432; TL: “Insel Nias”.

  **Distribution:** Nias Is. [Indonesia].

  ***L. neptunus lepida* (Hanafusa, 1994)**

  *Losaria neptunus lepida* (Hanafusa, 1994); Futao, **17**: 18, pl. 3, f. 13-14; TL: “Tanahmasa Is., Batu Islands, Indonesia”.

  **Distribution:** Tanahmasa Is. [Indonesia].

  ***L. neptunus eminens* (Hanafusa, 1990)**

  *Losaria neptunus eminens* (Hanafusa, 1990); Futao, **6**: 9, pl. 2, f. 5-7; TL: “Sipora Island, Kep. Mentawai, Indonesia”.

  **Distribution:** Sipora Is. [Indonesia].

  ***L. neptunus siborangitana***
**(Tsukada and Nishiyama, 1980)**

  *Losaria neptunus* ssp. *siborangitana* (Tsukada and Nishiyama, 1980); in Tsukada (Ed.), Butts SE Asian Islands, **1**: 259, pl. 52, f. 3-4; TL: “near Bandar Baru N. Sumatra”. [near Bandar Baru, Northern Sumatra, Indonesia].

    *Papilio neptunus* var. *Sumatrana* (Hagen, 1894); Dt. ent. Z. Iris, **7**(1): 21; TL: “Sumatra ... Vorbergen Delis” [Medan foothills, North Sumatra, Indonesia]. [Junior homonym of *Papilio cloanthus* var. *Sumatrana* (Hagen, 1894), chosen by Moonen [39] as the senior name of three homonyms introduced in the same work.]

     *Pachliopta neptunus muadder* Koçak and Kemal, 2000; Misc. Pap. Centre ent. Stud., **71**: 3 [unnecessary replacement name].

  **Distribution:** Northern Sumatra [Indonesia].

  ***L. neptunus padanganus* (Rothschild, 1908)**

  *Papilio neptunus padanganus* (Rothschild, 1908); Novit. zool., **15**(1): 165; TL: “West Sumatra: Padang and Padang Sidempoean”. [Padang and Padang Sidempuan, Western Sumatra, Indonesia].

  **Distribution:** Western Sumatra [Indonesia].

  ***L. neptunus doris***** (Rothschild, 1908)**

  *Papilio neptunus doris* (Rothschild, 1908); Novit. zool., **15**(1): 165; TL: “North Borneo”.

  **Distribution:** The island of Borneo.

  ***L. neptunus dacasini* (Schröder, 1976)**

  *Pachliopta neptunus* ssp. *dacasini* (Schröder, 1976); Ent. Z., **86**(24): 269; TL: “Philippinen, Palawan”.

  **Distribution:** Palawan Is. [Philippines].

  ***L. neptunus matbai***
**(Schröder and Treadaway, 1990)**

  *Losaria neptunus* ssp. *matbai* (Schröder and Treadaway), 1990; Ent. Z., **100**: 381; TL: “Philippinen, Sulu-Archipel, Tawitawi-Gruppe” [Tawitawi Islands, Sulu Archipelago, Philippines].

* ***Distribution:** Tawitawi Is. [Sulu Archipelago, southern Philippines].


***Losaria coon* (Fabricius, 1793)**


  ***L. coon palembanganus* (Rothschild, 1896)**

  *Papilio coon palembanganus* (Rothschild, 1896); Novit. zool., **3**(4): 421; TL: “Upper Musi River, Palembang district, Sumatra; 103° E. Long., 3° S. Lat.”.

  **Distribution:** Southern Sumatra [Indonesia].

  ***L. coon coon* (Fabricius, 1793)**

  *Papilio Coon* (Fabricius, 1793); Ent. syst., **3**(1): 10, no. 27; TL: “China” [loc. err. = Java].

  *   **Papilio Hypenor* (Godart, 1819); in: Latreille M. and Godart M., Encyc. méth., **9**(1): 65; TL: “Java”.

  **Distribution:** West Java [Indonesia].

  ***L. coon patianus* (Fruhstorfer, 1898)**

  *Papilio coon patianus* (Fruhstorfer, 1898); Soc. ent., **12**(23): 179; TL: “Pati in der Residenz Djapara, dem nördlichsten Teile von Central-Java” [Pati, northernmost part of Central Java].

  **Distribution:** Northern Central Java [Indonesia].


***L. coon sangkapurae* (Bollino and Sala, 1992)**


  *Atrophaneura coon sangkapurae* (Bollino and Sala, 1992); Trop. Lepid., **3**(2): 119, f. 1, 5-8; TL: “INDONESIA: Pulau Bawean” [Bawean Is., Indonesia].

  **Distribution:** Bawean Is. [Indonesia].


***Losaria rhodifer* (Butler, 1876)**


*Papilio rhodifer* (Butler, 1876); Ent. month. Mag. **13**(147): 57; TL: “Andaman Islands”.

  **Distribution:** Andaman Islands, [India].


***Losaria doubledayi* (Wallace, 1865) stat. rev.**


  ***L. doubledayi cacharensis* (Butler, 1885) comb. nov.**

  *Papilio cacharensis* (Butler, 1885); Ann. Mag. Nat. Hist. (Ser. 5), **16**: 344; TL: “Near Assam … Cachar” [Cachar, Assam, N.E. India].

  **Distribution:** Assam and Meghalaya, N.E. India.

  ***L. doubledayi putaoa* (Tytler, 1939) comb. rev.**

  *Polydorus doubledayi putaoa* (Tytler, 1939); J. Bomb. nat. Hist. Soc., **41**(2): 236; TL: “Putao, N.-E. Burma” [Putao, Kachin State, North Myanmar].

*  ***Distribution:** Upper Irrawaddy River valley, N. Myanmar.

  ***L. doubledayi sambilanga* (Doherty, 1886) comb. rev.**

  *Papilio doubledaii*[sic] var. *sambilanga* (Doherty, 1886); J. asiat. Soc. Bengal (Pt. II), **55**(3): 263; TL: “Great Nicobar”.

  **Distribution:** Nicobar Islands [India].

  ***L. doubledayi delianus* (Fruhstorfer, 1895) comb. rev.**

  *Papilio doubledayi* var. *delianus* (Fruhstorfer, 1895); Ent. Nachr., **21**(12/13): 196; TL: “Deli, Sumatra” [Medan, N. Sumatra, Indonesia].

  **Distribution:** Northeastern and northern Sumatra [Indonesia].

  ***L. doubledayi doubledayi* (Wallace, 1865)**

  *Papilio Doubledayi* (Wallace, 1865); Trans. linn. Soc. Lond., **25**: 42; TL: “Moulmein, Assam” [Mawlamyine, southern Myanmar and Assam, N. E. India].

  *   **Polydorus doubledayi merguia* (Tytler, 1939); J. Bomb. nat. Hist. Soc., **41**(2): 236; TL: “Victoria Pt., Mergui” [Myeik, Tanintharyi, southern Myanmar].

  **Distribution:** Malay Peninsula, southern Myanmar, Thailand, Laos, Cambodia, Vietnam.

  ***L. doubledayi insperatus* (Joicey and Talbot, 1921) comb. nov.**

  *Papilio coon insperatus* (Joicey and Talbot, 1921); Bull. Hill Mus. Witley, **1**(1): 168, pl. XIX, f. 2; TL: “Interior, Hainan”.

  **Distribution:** Hainan Is. [China].

## 4. Discussion

Our findings showed that *L. coon* and *L. doubledayi* are two distinct species within the genus *Losaria*. The molecular data suggested that *L. coon* is sister to *L. rhodifer*, which are then sister to *L. doubledayi*, as shown in Figure 2. Morphological analyses of genitalic structures and wing patterns are consistent with the molecular evidence, as shown in Figure 3, Figure 4, Figure 5, Figure 6, Figure 7 and Figure 8; therefore, we confirm the species status of *L. doubledayi*. Furthermore, our morphological analysis, based on the literature [2] and type photos of all recognised subspecies of *L. coon* and *L. doubledayi*, allowed us to update the subspecies list of each species, with *L. coon,* containing four subspecies, and *L. doubledayi,* containing six subspecies. As stated under Materials and Methods (Taxon sampling), the present study was unable to include all subspecies due to difficulties in obtaining specimens from India, northern Myanmar and some Indonesian islands, and consequently partial findings can be presented here for subspecies. Future research must address this point to form a better understanding of the subspecies divergence, diversity, and distribution pattern when those taxa become accessible.

The divergence between *L. coon coon* and *L. coon sangkapurae* is limited, as shown in Table 2 and Table 3 and Figure 2, despite the two subspecies being isolated on different islands. Geographic proximity and recent separation between Sumatra, Java, and those offshore small islands might explain such limited divergence [40,41,42,43]. Based on the distribution pattern and morphological characters of the other two subspecies [2], namely ssp. *palembanganus* and ssp. *patianus*, it is likely that the divergence between them would also be limited.

In comparison, the divergence within *L. doubledayi* is greater. The phylogenetic tree, genetic distances and molecular species delimitation results showed that even the populations from the Malay Peninsula and upper Indochina are different, let alone the insular ssp. *insperatus* from Hainan, as shown in Table 3 and Figure 2. Our examination of a large series of specimens, type photos, and literature records also showed morphological differences among these populations/subspecies. For instance, populations from upper Indochina are usually smaller in size and possess larger hindwing white discal patches compared to populations from the Malay Peninsula, and the population from South Vietnam (samples LCN018 and LCN019 in our dataset, other examined specimens, and also in the cited literature) possesses unique ochreous abdomens, rather than crimson ones [44]. Similarly, photos of live specimens of ssp. *cacharensis,* plus type photos and the description of ssp. *putaoa* also showed constant morphological differences compared to other populations [14,45]. Future research should address the question of the population divergence of *L. doubledayi* across its distribution range.

The molecular and morphological divergence within *L. doubledayi* across its range might be attributed to two factors. The first factor lies in the more heterogeneous natural environment (e.g., terrain topology, climate, and landscape). The valley habitat in North Myanmar and Northeast India, separated by mountains, where ssp. *putaoa* and ssp. *cacharensis,* respectively, occur, is a typical example of divergence. Such extreme altitude shifts restricted populations within narrow ranges (mostly river valleys with broadleaf forest patches), and they subsequently became distinct evolutionary entities (subspecies). Similarly, the southern–central savannah plain in Thailand and Cambodia formed an effective barrier due to the lack of suitable forests for the larval foodplants. The second factor is the larval foodplants for *Losaria*, which constitute a handful of species in the genus *Thottea* (family Aristolochiaceae) [46]. *Thottea* is a genus of lowland forest-dwelling shrub species with high endemism; habitats without dense forests would be unsuitable for *Losaria* [47,48]. The superimposition of both factors resulted in spatially fragmented suitable habitats, which eventually lead to divergence within *L. doubledayi*.

Given the separation of *L. coon* and *L. doubledayi*, conservation strategy and threat evaluation of the original *L. coon* should be reassessed. According to Fernando, Jangid, Chowdhury, Kehimar, Lo, and Moonen [12], *L. coon* and *L. doubledayi* were considered as a single wide-ranging species, with an IUCN rank of Least Concern (LC). After its work separating these two species, *L. coon* now becomes a narrow-ranged species that only occurs in several Indonesian islands, including southern Sumatra, Java, and Bawean, as shown in Figure 1. Hence, conservation and assessment strategies must be updated accordingly to ascertain the current population status. Furthermore, as a separate species, although *L. doubledayi* occupies a much wider range, it still possesses more localised subspecies, even within continental Indochina, as shown in Figure 1 and Figure 2. Each population represents a distinct evolutionary pool, while some of them only occupy a much smaller geographic range than others (e.g., ssp. *cacharensis* and ssp. *putaoa*). Since anthropogenic disturbance to most of its distribution range remains active to date [49,50,51,52,53], we think that the population status, vulnerability, and threat factors of each subspecies of *L. doubledayi* must be re-evaluated for future conservation.

## 5. Conclusions

The present study used DNA barcode data and morphological characters to unveil the relationship of the yellow and red marked *Losaria coon* across its distribution range. Our analyses support the separation of *L. coon* (yellow marked) and *L. doubledayi* (red marked) as two distinct species. Our findings also updated the geographic ranges of the two species: with *L. coon* restricted to southern Sumatra, Java, and Bawean Island, while *L. doubledayi* occurs widely in regions from North India to northern Sumatra, including Hainan and Nicobar Islands. The separation effectively made *L. coon* a narrow-ranged species which may require more conservation attention in the future, while that for *L. doubledayi* must also be revised due to its higher subspecies diversity reflecting multiple evolutionary pools associated with larval foodplant restriction and landscape heterogeneity.

## Figures and Tables

**Figure 1 insects-11-00392-f001:**
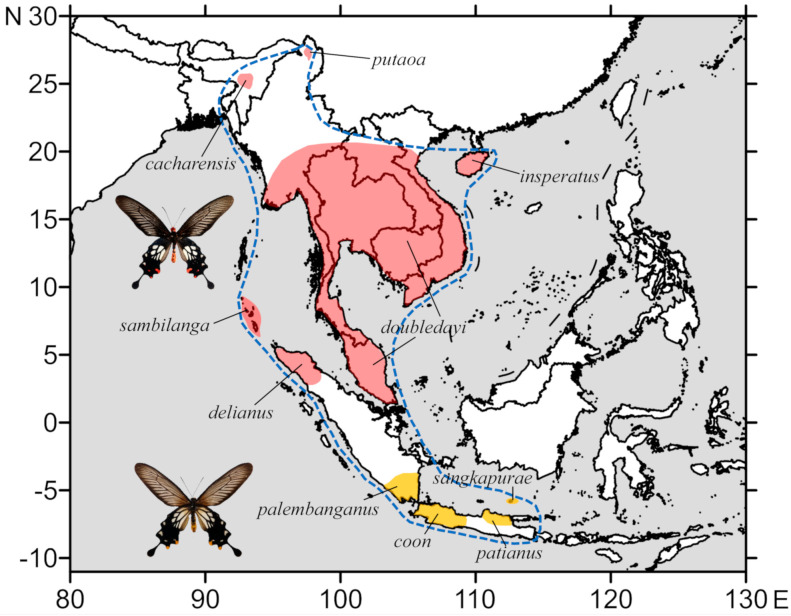
Distribution of the yellow and red marked *Losaria coon* (Fabricius, 1793), with currently accepted subspecies delimitation; the blue dash line encircles the tentative distribution range of *L. coon*.

**Figure 2 insects-11-00392-f002:**
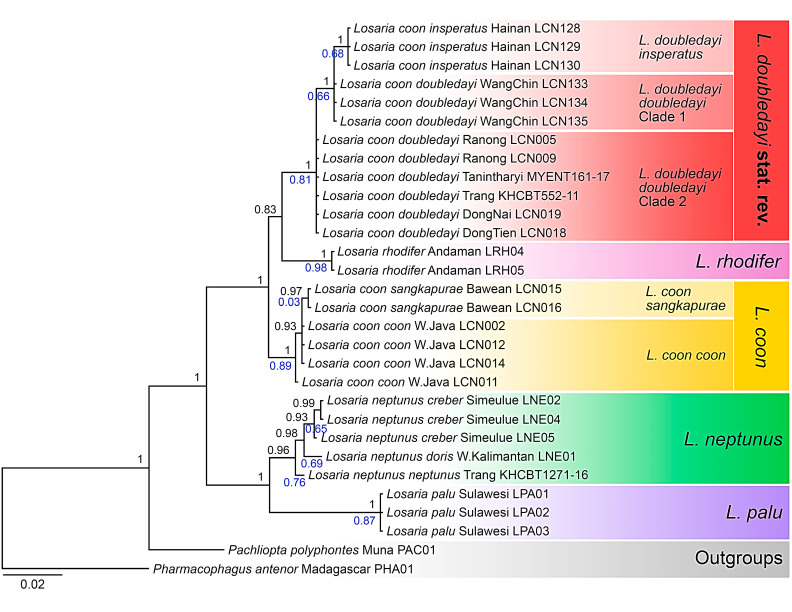
The Bayesian phylogenetic tree of genus *Losaria* (Moore, 1902) with *Pharmacophagus antenor* and *Pachliopta polyphontes* as outgroups. The branch tip labels of taxon names follow the current taxonomy, while the names in red colour blocks reflect changes proposed in this study. Black values at nodes indicate the posterior probability, and the blue values indicate the probabilities for *Losaria* taxa identified by the Bayesian–Poisson tree process model (bPTP).

**Figure 3 insects-11-00392-f003:**
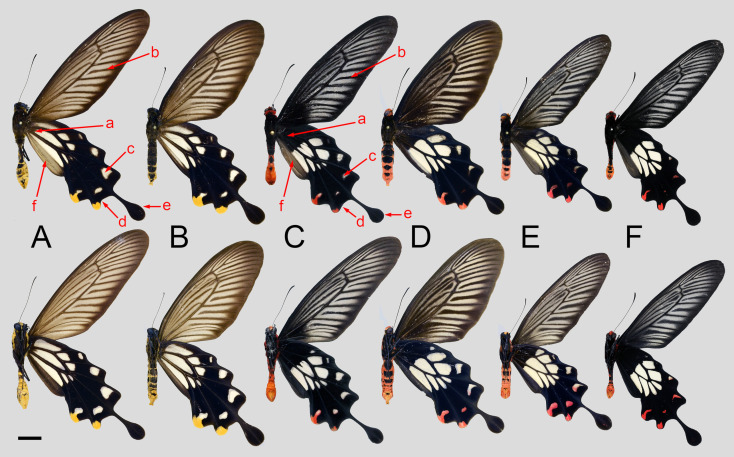
Morphological comparison of *Losaria coon* and *L. doubledayi* A: *L. coon coon*, West Java, Indonesia, male ♂; B: ditto, female ♀; C: *L. doubledayi doubledayi*, Ranong, Thailand, male ♂; D: *L. doubledayi doubledayi*, Perak, Malaysia, female ♀; E: *L. doubledayi doubledayi*, Wang Chin, Thailand, male ♂; F: *L. doubledayi insperatus*, Hainan, China. Scale bar = 10 mm.

**Figure 4 insects-11-00392-f004:**
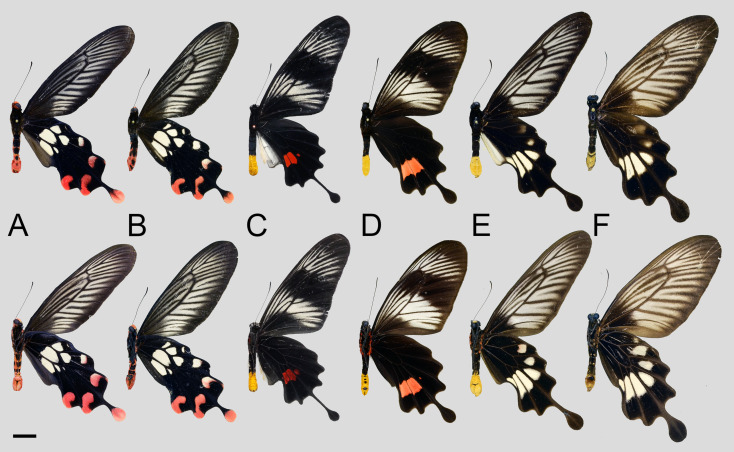
Morphological comparison of *Losaria rhodifer* (A: male ♂, B: female ♀), *L. neptunus* (C: male ♂, D: female ♀), and *L. palu* (E: male ♂, F: female ♀). Upper side on the first row, underside on the second row; scale bar = 10 mm.

**Figure 5 insects-11-00392-f005:**
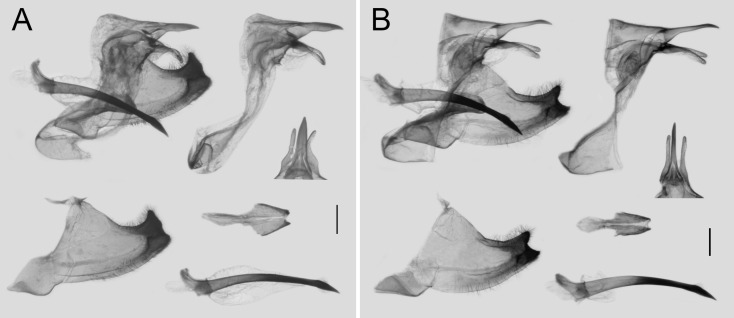
Comparison of male genitalia of *Losaria coon* (**A**) and *L. doubledayi* (**B**); scale bars = 1.0 mm.

**Figure 6 insects-11-00392-f006:**
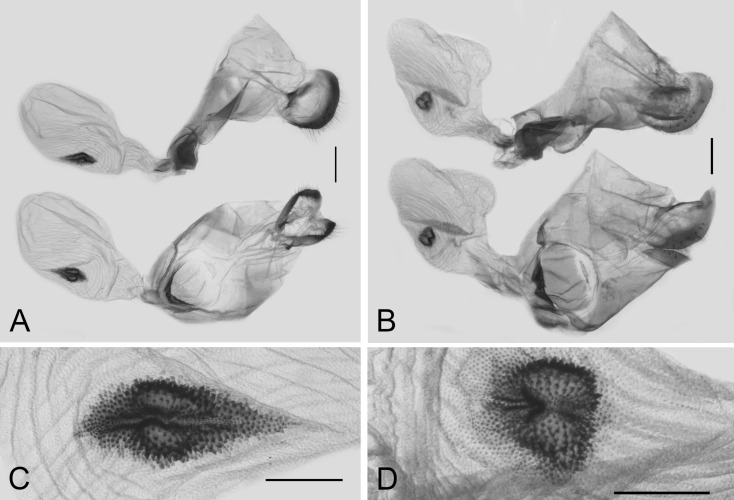
Comparison of female genitalia of *Losaria coon coon* (**A**) and *L. doubledayi* (**B**), scale bars = 1.0 mm; with enlarged signum (**C**) and (**D**), scale bars = 0.5 mm.

**Figure 7 insects-11-00392-f007:**
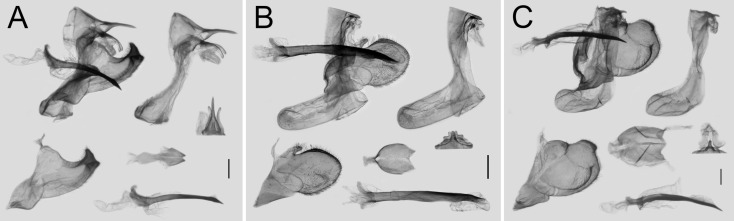
Male genitalia of *Losaria rhodifer* (**A**), *L. neptunus* (**B**), and *L. palu* (**C**); scale bars = 1.0 mm.

**Figure 8 insects-11-00392-f008:**
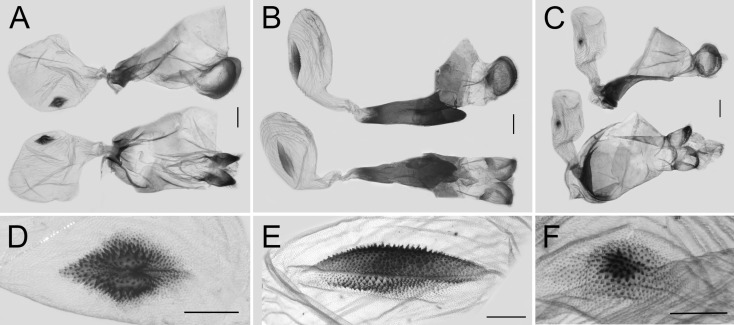
Female genitalia of *Losaria rhodifer* (**A**), *L. neptunus* (**B**), and *L. palu* (**C**), scale bars = 1.0 mm; with enlarged signum (**D**–**F**), scale bars = 0.5 mm.

**Table 1 insects-11-00392-t001:** Sampling information and GenBank/BOLD accession numbers of the *Losaria* species and outgroups used in this study. The taxon names follow current taxonomy, mentioned above.

Taxon (Sample Code)	Locality	Collecting Date	Accession No.
*Losaria coon coon* (LCN002)	West Java, Indonesia	2018-I	MT417883
*Losaria coon coon* (LCN011)	West Java, Indonesia	2018-I	MT417884
*Losaria coon coon* (LCN012)	West Java, Indonesia	2018-I	MT417883
*Losaria coon coon* (LCN014)	West Java, Indonesia	2018-I	MT417883
*Losaria coon sangkapurae* (LCN015)	Bawean Is., Indonesia	2018-II	MT417885
*Losaria coon sangkapurae* (LCN016)	Bawean Is., Indonesia	2018-II	MT417885
*Losaria coon doubledayi* (MYENT161-17)	Lenya, Tanintharyi, Myanmar	2015-V-29	MYENT161-17
*Losaria coon doubledayi* (KHCBT552-11)	Khao Chong, Trang, Thailand	2010-XII-26	KHCBT552-11
*Losaria coon doubledayi* (LCN005)	Ranong, Thailand	2018-II	MT417886
*Losaria coon doubledayi* (LCN009)	Ranong, Thailand	2018-II	MT417886
*Losaria coon doubledayi* (LCN018)	Dong Tien, Binh Thuan, Vietnam	2018-VII	MT417886
*Losaria coon doubledayi* (LCN019)	Thac Mai, Dong Nai, Vietnam	2018-VIII	MT417886
*Losaria coon doubledayi* (LCN133)	Wang Chin, Phrae, Thailand	2017-XI-11	MT417888
*Losaria coon doubledayi* (LCN134)	Wang Chin, Phrae, Thailand	2017-XI-11	MT417888
*Losaria coon doubledayi* (LCN135)	Wang Chin, Phrae, Thailand	2017-XI-11	MT417888
*Losaria coon insperatus* (LCN128)	Sanya, Hainan Is., China	2018-V	MT417887
*Losaria coon insperatus* (LCN129)	Sanya, Hainan Is., China	2018-V	MT417887
*Losaria coon insperatus* (LCN130)	Sanya, Hainan Is., China	2018-V	MT417887
*Losaria rhodifer* (LRH04)	Andaman Is., India	2014-X	MT417889
*Losaria rhodifer* (LRH05)	Andaman Is., India	1962-III	MT417890
*Losaria neptunus neptunus* (KHCBT1271-16)	Khao Chong, Trang, Thailand	2014-V-2	KHCBT1271-16
*Losaria neptunus doris* (LNE01)	West Kalimantan, Indonesia	2018-I	MT417891
*Losaria neptunus creber* (LNE02)	Simeulue Is., Indonesia	2010-VIII	MT417892
*Losaria neptunus creber* (LNE04)	Simeulue Is., Indonesia	2010-VIII	MT417893
*Losaria neptunus creber* (LNE05)	Simeulue Is., Indonesia	2010-VIII	MT417894
*Losaria palu* (LPA01)	Donggala, Sulawesi, Indonesia	2017-X	MT417895
*Losaria palu* (LPA02)	Donggala, Sulawesi, Indonesia	2018-II	MT417895
*Losaria palu* (LPA03)	Donggala, Sulawesi, Indonesia	2018-II	MT417895
*Pachliopta polyphontes* (PAC01)	Muna Is., Indonesia	2017-XII	MT417896
*Pharmacophagus antenor* (PHA01)	Madagascar	2016-VI	MT417897

**Table 2 insects-11-00392-t002:** Monophylizer assessment of the *Losaria* species and subspecies used in this study.

Taxon	Assessment	Tanglees
1. *L. coon*	monophyletic	—
1a. *L. coon coon*	paraphyletic	*L. coon sangkapurae*
1b. *L. coon sangkapurae*	monophyletic	—
2. *L. doubledayi*	monophyletic	—
2a. *L. doubledayi doubledayi* 1	paraphyletic	*L. doubledayi doubledayi* 2;*L. doubledayi insperatus*
2b. *L. doubledayi doubledayi* 2	paraphyletic	*L. doubledayi insperatus*
2c. *L. doubledayi insperatus*	monophyletic	—
3. *L. rhodifer*	monophyletic	—
4. *L. neptunus*	monophyletic	—
5. *L. palu*	monophyletic	

**Table 3 insects-11-00392-t003:** The Kimura two-parameter distances (in percentages) between all taxa of genus *Losaria* (Moore, 1902), with species and subspecies identified as in the Bayesian phylogenetic tree, as shown in Figure 2.

	1a	1b	2a	2b	2c	3	4	5
1a. *L. coon coon*								
1b. *L. coon sangkapurae*	0.19							
2a. *L. doubledayi doubledayi* 1	3.08	3.28						
2b. *L. doubledayi doubledayi* 2	2.44	2.64	0.61					
2c. *L. doubledayi insperatus*	3.57	3.77	0.46	1.08				
3. *L. rhodifer*	3.17	3.35	3.35	3.00	3.35			
4. *L. neptunus*	6.11	6.32	6.76	6.32	7.03	6.47		
5. *L. palu*	7.83	8.04	8.92	8.40	9.28	9.61	5.19

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
