# Peer review of "Are the Yellow and Red Marked Club-Tail Losaria coon the Same Species?"

_insects, 2020, doi:10.3390/insects11060392_

Round 1

Reviewer 1 Report

The manuscript by Xu et al. entitled “Are the Yellow and Red Marked Losaria coon the Same Species?” is a valuable work that through an integrative taxonomy approach, including morphological and molecular evidence, aims to define if yellow and red marked Losaria coon individuals belong to the same species. In my opinion, the aim of this research is of interest, but more analyses are needed for clearly defining the status of analyzed taxa. Authors inferred a single marker Bayesian tree and used it to support the species status of the taxa based on molecular data, but the analysis of a phylogenetic tree is not enough for defining if the taxa are two separated species. In this case, molecular species delimitation analyses using almost one molecular species delimitation method for single locus data would give more reliable evidence for defining the status of the taxa (similar case in Plewa et al., 2018 Arthropod Systematics & Phylogeny 76: 123-135). Moreover, even if images of specimens and morphological characters details are high quality and well done, morphological differences among Losaria coon and Losaria doubledayi are not so precisely described. Differences in characters between species are described as “shorter than” “broader than” but any range of length/dimension for each species is given. Moreover, in studies like this in which morphological differences between species are analyzed for defining if they are different taxa, geometric morphometric analysis is useful to statistically proof their separation (Montagna et al., 2016 Zoological Journal of Linnean Society 179: 92-109). Finally, within the manuscript some other points, listed below, should be modified or clarified.

Abstract:

The method used for achieving the study aim is not well explained in the abstract, in particular the sentence at lines 18-21 is confused.

Materials and Methods

Line 81: Since sampling basically is not described, I suggest to change the title of this section with a more appropriate one.

Line 84: even if the specimens analyzed in this study come from authors private collections, some information about how they were collected and conserved should be given, especially because obtaining good results from a DNA extraction performed on dry conserved insects is dependent from how they were collected and stored.

Line 99: In my opinion this title is not so appropriate, maybe you could write something like “DNA extraction and PCR” or similar.

Line 101: what is “protease butter”?

Lines 106-109: please clarify better this part, it seems you are listing what the kit contains and not your PCR mix (“kit that contained”).

Line 110: I suggest to use only one acronym for cytochrome oxidase subunit I.

Line 117: Could you please specify better how you proof raw sequences through Clustal W?

Line 118: Did you excluded sequences containing double peaks also when you had only few of them in an electropherogram?

Line 125: Did you pruned identical haplotypes before tree inference? Observing the obtained tree it seems some zero length branches are present.

Line 133: I suggest to include in this paragraph also some information about how specimens were morphologically identified, for example citing the specialistic literature adopted.

Results:

Lines 155-162: Some L. coon and L. doubledayi subspecies are not monophyletic.

Lines 163-172: in this paragraph “smaller” and “greater” should be substituted with the observed nucleotide distance values. Moreover, in my opinion, to underline the nucleotide divergence between L. coon and L. doubledayi is more important that showing differences among clades within these species. For this reason, I suggest to include between L. coon and L. doubledayi nucleotide distances in table 2.

Lines 173-176: In my opinion, if you are adopting an integrative taxonomic approach (accounting for morphological and molecular evidence) to defining the status of L. coon and L. doubledayi you should analyze both evidence before to draw any conclusion.

Line 188: I suggest to use taxa here instead of species, in accordance with what I suggested in the previous sentence.

Discussion:

Line 378: In my opinion, the one you reported is not the reason why you obtained 0.74 PP value, could you add some references demonstrating low PP values are related to low sample size?

Reviewer 2 Report

This manuscript addresses the problem of species delimitation in the butterflies of the genus Losaria leading to the conclusion that the allopatric taxa L.doubledayi (red marked) and L.coon (yellow marked) are separate species.

This conclusion, although advisable and instrumental for conservation purposes, is based on limited molecular (only the barcode sequence of the mitochondrial COI gene) and morphological evidence.

This manuscript is on the whole clear and well documented from a systematic and biogeographical point of view, it deals with an appealing genus of Oriental butterflies and could be published on Insects, but suffers from incompleteness and approximation on some points.

1) Taxon sample: both molecular and morphological analyses were carried out only on a subset of the geographical races known within the genus. Variation at the population level is obviously crucial to understand boundaries at the species level. Incomplete taxon sampling obviously weakens taxonomic conclusions at the species level.

2) The main criterion used at the molecular level was that of the distances between the yellow and red marked coon, with overall range falling within 2.44% to 3.17%. This was compared simplistically with the proposed and in any case arbitrary, barcoding gap of 2%. The Authors are invited to explore methods like Automatic Barcode Gap Discovery (ABGD), to detect the so-called “barcode gap” in the distribution of pairwise distances between sequences (Puillandre et al. 2012: Molecular Ecology 21: 1864-1877). Even adopting a phylogenetic concept of species, the simple analysis based on the current Bayesian phylogenetic tree appears not satisfactory because of the lack of clear monophyletic structure of the putative species. Authors should at least attempt exploring methods based on the multispecies coalescent model (MSC) such as the general mixed Yule coalescent (GMYC, Pons et al. 2006: Systematic Biology 55: 595–609) or a given phylogenetic input tree by means of the Poisson tree processes (PTP, Zhang et al. 2013: Bioinformatics 29: 2869-2876).

3) The morphological analysis, especially of the genitalia, is overall convincing, yet it is conducted in a rather typological form. In fact, there is no hint or circumstantial evidence on the level of intra-population morphological variation such as to allow a statistical evaluation of the differences. Only a morphological constancy within the 'species' is declared. These aspects are not even touched on either in the Materials and Methods section nor in the Discussion.

Since these yellow and red marked Losaria are typically allopatric taxa, where the biological concept of species finds greater application difficulties, the integrated, multi-dimensional approach attempted by the authors would deserve to be more strongly supported on a statistical basis!

Reviewer 3 Report

General comments

This is a nicely written study that through morphological, molecular and phylogenetic evidence supports the separation of Losaria coon and L. doubledayi into two distinct species. While the evidence is adequate and robust, convincing thus the reader on the validity of the output, I am wondering if the authors could go deeper and try to link the smaller body size they found in the upper Indochina. This is indeed an interesting finding and can be associated to the shrinking body size of biodiversity that results from climate change. A shrinking body size does affect ecosystem functioning which in turn is crucial from a conservation and ecological point of view.

Minor points

-Please replace the title with another one, more informative.

-Please consider deleting the name Jordan in Line 16.

-Line 21: I am not sure what stat. rev. stands for; do you mean revised statuses (rev. stat.)?

Round 2

Reviewer 1 Report

The manuscript entitled “Are the Yellow and Red Marked Losaria coon the Same Species” has been substantially improved after the revision. In any case, some points that should be better clarified remain. The most important regards the new performed species delimitation analyses.

Hereafter I list my suggestions in detail:

Lines 80-97: I appreciated the authors followed the suggestion of adding information on specimens sampling, but now this paragraph results a bit confused. I suggest to better separate the information about specimens collected from authors used for morphological analysis and then for DNA extraction from the part about specimens used only for molecular analyses, mined sequences and outgroup choice.

Lines 110-113: Remember that LCO-HCO primers pair amplifies only a region of Cox1 gene, that in the last decades has been named as “barcode region”.

Line 120: I suggest to substitute the word “showed” with “were present”.

Lines 137-145: It seems you are applying ABGD to the BI tree, but this is a distance-based method. I suggest to clarify this point. In addition, just to be sure, did you applied ABGD setting a Pmax of 0.01 or of 0.1 (the range of P value was set from 0.001 to 0.01)?

Lines 177-186: Also here it seems you have applied ABGD to the tree. Moreover, you are saying that bPTP and ABGD showed similar results and recognized 9 entities, but not showing anywhere ABGD results. I suggest to report ABGD results in the text more explicitly and also to add them to figure 2. For understanding and comparing better the results of species delimitation analyses is common to draw vertical lines in a side of the tree showing the entities delimited by the different delimitation methods, I think this could be useful also in your case. Finally, I suggest to define these analyses “molecular species delimitation analyses” instead of “taxon delimitation analyses”.

Line 411: I suggest to specify you are speaking about materials and methods when you mention “Taxon sampling”.

Discussion: if the results of molecular species delimitation are slightly different you should discuss them.

Reviewer 2 Report

In their cover letter, the Authors responded accurately to the objections raised on the first copy of their manuscript.

In particular, they convincingly discussed the difficulties of improving their research with respect to points 1 (taxon sampling) and 3 (morphological analysis).

However, I believed point 2 to be decisive with respect to the arbitrariness of considering a certain value as a ‘barcoding gap’.  On this point the authors have provided a convincing answer by developing, as I suggested, new analyses (according to two methods: the bPTP and the ABGD to the tree) and discussion to support their major point i.e.: the identification of a barcoding gap among Losaria species as the main conclusion of their research.

On this basis I recommend publishing the manuscript in its present form.

Author Response

Reply: Thank you for the confirmation of the revision and edits, as well as your recommendation for publication.